

# The impact of FASTQ and alignment read order on structural variant calling from long-read sequencing data

Kyle J. Lesack[1,2] and James D. Wasmuth[1,2]

[1] Faculty of Veterinary Medicine, University of Calgary, Calgary, Alberta, Canada
[2] Host-Parasite Interactions Research Training Network, University of Calgary, Calgary, Alberta, Canada

## ABSTRACT

**Background:** Structural variant (SV) calling from DNA sequencing data has been challenging due to several factors, including the ambiguity of short-read alignments, multiple complex SVs in the same genomic region, and the lack of "truth" datasets for benchmarking. Additionally, caller choice, parameter settings, and alignment method are known to affect SV calling. However, the impact of FASTQ read order on SV calling has not been explored for long-read data.
**Results:** Here, we used PacBio DNA sequencing data from 15 *Caenorhabditis elegans* strains and four *Arabidopsis thaliana* ecotypes to evaluate the sensitivity of different SV callers on FASTQ read order. Comparisons of variant call format files generated from the original and permutated FASTQ files demonstrated that the order of input data affected the SVs predicted by each caller. In particular, pbsv was highly sensitive to the order of the input data, especially at the highest depths where over 70% of the SV calls generated from pairs of differently ordered FASTQ files were in disagreement. These demonstrate that read order sensitivity is a complex, multifactorial process, as the differences observed both within and between species varied considerably according to the specific combination of aligner, SV caller, and sequencing depth. In addition to the SV callers being sensitive to the input data order, the SAMtools alignment sorting algorithm was identified as a source of variability following read order randomization.
**Conclusion:** The results of this study highlight the sensitivity of SV calling on the order of reads encoded in FASTQ files, which has not been recognized in long-read approaches. These findings have implications for the replication of SV studies and the development of consistent SV calling protocols. Our study suggests that researchers should pay attention to the input order sensitivity of read alignment sorting methods when analyzing long-read sequencing data for SV calling, as mitigating a source of variability could facilitate future replication work. These results also raise important questions surrounding the relationship between SV caller read order sensitivity and tool performance. Therefore, tool developers should also consider input order sensitivity as a potential source of variability during the development and benchmarking of new and improved methods for SV calling.

Corresponding author
James D. Wasmuth,
jwasmuth@ucalgary.ca

## INTRODUCTION

Structural variants (SVs) describe a broad category of large chromosome alterations, often defined at greater than 100 bp, that include deletions, insertions, inversions, translocations, and duplications. SVs are a major source of genetic diversity that often account for more nucleotide level differences within and between species compared to single nucleotide variants (*Perry, 2009*; *Pang et al., 2010*; *Long et al., 2018*; *Catanach et al., 2019*; *Stuart et al., 2023*). Major phenotypic consequences are associated with SVs, especially those that disrupt genes or modulate gene expression (*Nowakowska, 2017*). Notably, hundreds of microdeletions and microduplication syndromes (*Wetzel & Darbro, 2022*) are associated with copy number variants, which contribute to approximately 10% of rare disorders (*Truty et al., 2019*). While SVs are often deleterious, they also play key evolutionary roles as a source of novel genes and adaptive phenotypes (*Radke & Lee, 2015*). In particular, gene duplications are described as a mechanism that provides the raw material for natural selection to act upon (*McGrath et al., 2014*). Theoretically, this may lead to adaptive changes in a process known as neofunctionalization, where one of the duplicated genes may diverge and acquire a novel function (*Zhang, 2003*; *Hurles, 2004*). Alternatively, gene duplication may result in sub-functionalization, where the ancestral function is partitioned between the two duplicates (*Hurles, 2004*). Natural selection is predicted to act on these two duplicated genes separately, resulting in further fitness benefits (*Zhang, 2003*). Increases in gene copy numbers may also contribute to adaptive evolution if increases in gene dosage provide a selective advantage (*Katju & Bergthorsson, 2013*). Copy number neutral changes can also play major roles in genome evolution and adaptation. Inversions, as an example, have been theorized to facilitate local adaptation and speciation. For example, locally adapted alleles can be preserved from gene flow through the suppression of recombination in the inverted region (*Huang & Rieseberg, 2020*). Within a population, the suppression of recombination can lead to adaptive divergence, which may eventually result in speciation (*Huang & Rieseberg, 2020*).

The detection of SVs from DNA sequencing data has proved to be challenging and the barriers to accurate and comprehensive SV prediction are multifactorial. Many variants span genomic regions that exceed the read sizes generated by high-throughput sequencing platforms, which may hinder accurate mapping to a reference genome (*Sedlazeck et al., 2018*). Because SVs, such as deletions and duplications, typically arise in low complexity regions, this problem is compounded by ambiguous mappings that are inherent to short-read alignments (*Treangen & Salzberg, 2012*). Multiple SVs may also occur in the same genomic region (*Carvalho & Lupski, 2016*; *Lin & Gokcumen, 2019*; *Cook et al., 2020*), leading to complex rearrangements that are difficult to resolve computationally (*Weckselblatt & Rudd, 2015*).

Considerable work has gone into improving SV calling methods, but the limited availability of "truth" datasets with known variants has limited benchmarking efforts (*Leung et al., 2015*; *Mahmoud et al., 2019*; *Chen et al., 2023*). Several sources of variability have also been identified that contribute further difficulties for the development of

consistent SV calling protocols. While differences in library preparation (*Guan & Sung, 2016*) or sequencing platform can affect the predicted SVs, considerable disparities between the call sets generated by different sequencing centers has been observed when using the same protocols (*Khayat et al., 2021*). On the computational side, caller choice, parameter settings, and alignment method are known to affect SV calling (*Lesack et al., 2022*; *Liu et al., 2022a*). For short-read data, how software handles ambiguous read-to-genome mappings is a surprising and significant source of variation in SV identification; changing the order of the reads in the FASTQ file led to changes in predicted SVs (*Firtina & Alkan, 2016*). These discrepancies were as high as 25% for certain callers, raising the possibility that the random nature of FASTQ read order could have a substantial impact on replication work. Without knowing ground truth, it is not possible to quantify the exact relationship between these SV call discrepancies and caller accuracy. Nonetheless, high sensitivity to read order reflects poor performance for certain metrics. Theoretically, a given caller may have a high true positive rate but may also generate considerably different predictions when the FASTQ file read order is changed. This would be consistent with the caller having high precision, but low recall, as the call set differences would include many true positives that are members of one call set but not the other. However, a caller with a high false positive rate could conceivably have a high recall, while also exhibiting high read order sensitivity. Here, most of the call set differences would correspond to false positives generated from a specific read order that were not generated in the other. An interesting question would be: Are the calls that differ following the randomization of read order biased towards being false positives? If any biases are found to exist, read order permutation could be used to improve the caller performance.

It is unclear if FASTQ read order should also be a consideration for SV calling from long-read data. To evaluate the FASTQ read order sensitivity of long-read callers, we used PacBio DNA sequencing data from 15 separate *Caenorhabditis elegans* strains and four *Arabidopsis thaliana* ecotypes. We generated FASTQ files with permutated read order from the original files and evaluated the differences between the SVs predicted using the initial and randomized data. Although each caller was found to be deterministic, the order of reads provided to each caller had an impact on the predicted SVs. Several factors were identified that contributed to the sensitivity of different SV callers on the order of FASTQ file reads. These results bring attention to a largely unrecognized factor that may affect the inferences made from structural variant studies.

## METHODS

Portions of this text were previously published as part of a preprint https://www.biorxiv.org/content/10.1101/2023.03.27.534439v1.full.

Snakemake (v7.9.0) was used to manage the individual scripts used for read order permutation, read alignment, subsampling, SV prediction, and statistical summary (*Köster & Rahmann, 2012*).

## DNA sequencing datasets and read order randomization

PacBio sequencing data were obtained for 14 *C. elegans* strains (DL238, ECA36, ECA396, EG4725, JU1400, JU2526, JU2600, JU310, MY2147, MY2693, NIC2, NIC526, QX1794, and XZ1516) from the *Caenorhabditis elegans* Natural Diversity Resource (CeNDR) database (*Cook et al., 2017*) and one sequencing run of the reference strain, N2 (SRA accession = DRR142768). PacBio sequencing data were obtained for four *A. thaliana* ecotypes (1254, 6021, 6024, and 9470; BioProject accession PRJNA779205) from the 1,001 Genomes Project (*Weigel & Mott, 2009*; *Jaegle et al., 2023*).

For each sequencing run, we created five permutated FASTQ files with randomized orders of the original reads. Initially, the BBTools (v39.00) shuffle.sh and shuffle2.sh scripts were used to randomize the FASTQ sequence order (*Bushnell, 2022*). However, in addition to randomizing the sequence order, the FASTQ files generated by both scripts contained changes in the Phred scores. Specifically, Phred scores encoded as "!" in the original FASTQ files were changed to "#" in the permutated versions. Therefore, the permutated FASTQ files were created using the seq-shuf script (*Hackl, 2023*). A comparison between the original and permutated FASTQ files indicated that only the order of sequences was altered using seq-shuf.

## Sequence alignment, subsampling, and structural variant prediction

Alignments for each FASTQ file were created using pbmm2 (v1.12.0) (*Pacific Biosciences, 2023*), Minimap2 (v2.26) (*Li, 2018a*), and NGMLR (v0.2.7) (*Sedlazeck et al., 2018*). The BAM files were then sorted using SAMtools (v1.9) (*Danecek et al., 2021*) and Picard (v2.27.5) (*Broad Institute, 2022*). SAMtools v.1.9 was used to sort the BAM files, as the CIGAR strings generated from these long-read alignments generated errors with newer releases of the tool. We do not anticipate that using a newer version would have affected the results, as the same sorting criteria, based on the leftmost genomic coordinate, was described in the documentation for SAMtools v.1.9 and subsequent releases. To identify any changes that resulted from read order randomization, the sorted BAM files generated from the original and permutated FASTQ files were compared with each other. While different BAM file read orders were observed in those sorted using SAMtools, the alignments contained identical information in the original and randomized versions. The Picard-sorted BAM files from the original and permuted FASTQ files were found to be identical. Identical output files were generated when the read alignment and sorting steps were repeated a second time using identical input data, indicating that these steps were deterministic.

SVs were predicted from the original and permutated datasets using the default parameters for three tools: pbsv (v2.9.0) (*Pacific Biosciences, 2022*), Sniffles (v2.2.0) (*Sedlazeck et al., 2018*), and SVIM (v2.0.0) (*Heller & Vingron, 2019*). Variant calls below 100bp were filtered out in order to limit the analysis to structural variants. We evaluated if each caller was deterministic by calling SVs twice using each alignment file. Apart from timestamps, the variant call format (VCF) files were identical to those from the original analysis. Given that the exact same sets of SV calls were predicted from a total of 342 BAM

files, we considered each caller to be deterministic (*i.e.*, the same results are obtained given identical input data).

Each FASTQ file was subsampled to 20X sequencing depth using Seqtk (v1.3) (*Li, 2018b*) prior to variant calling in both *C. elegans* and *A. thaliana*. To evaluate the impact of depth on FASTQ read order sensitivity, variant calling was performed for *C. elegans* at four subsampled depths (10X, 20X, 40X, and 60X), as well as the full depth of sequencing (Minimap2 median depth = 145.602; NGMLR median depth = 135.136; pbmm2 median depth = 136.831).

## Comparison of predicted structural variants

For each strain or ecotype, we compared the predictions generated from the original and permuted FASTQ files from the same aligner. The SV call sets generated from each caller included breakends (BND), deletions (DEL), tandem duplications (DUP), insertions (INS), and inversions (INV). Only the SVIM comparisons included interspersed duplications (DUP:I), as no other tool supported the identification of this SV type. Two approaches were used to assess the impact of read order randomization on the predicted structural variants: (1) VCF-level differences and (2) coordinate-level differences. At the VCF-level, the SV calls contained in the VCF files created from the original and permuted FASTQ files were compared using all VCF fields, except for the variant ID. The variant ID field was excluded because it only describes an arbitrary name assigned to the SV call from its order in the VCF file. Because an analyst may only be concerned with the locations of SVs that pass quality filtering, a less strict analysis was also performed that limited comparisons to the variant type, filter, chromosome, start coordinate, and end coordinate (not applicable to BND or INS calls).

For each comparison, the symmetric difference (the union of the SVs predicted from the original and permuted FASTQ file, but not in their intersection) and Jaccard distance (the ratio of the symmetric difference to the union) was used to quantify the differences that occurred following read order randomization. The differences and proportions described for each strain or ecotype are represented using the mean ± population standard deviation (SD). The impact of the alignment method and sequencing depth on FASTQ read order sensitivity are described as the global means of the differences and distances from all strains or ecotypes in each species. The values included in the figures indicate the mean proportion of different calls that resulted from read order permutation.

## RESULTS

### Structural variant calling is affected by FASTQ read order

Comparisons of VCF files generated from the FASTQ files with the original and randomized read orders demonstrated that the order of input data impacts SV prediction. For both *C. elegans* and *A. thaliana*, the overall differences between the SV call sets generated from the original and randomized FASTQ files were highest for SVIM, followed by pbsv, and Sniffles (Table 1). Within *C. elegans* and *A. thaliana*, the same callers generated the most differences for breakends (pbsv), tandem duplications (pbsv), insertions (SVIM), and inversions (pbsv). For deletions, pbsv and SVIM accounted for the

**Table 1 Average differences between the SV call sets generated from the original and randomized FASTQ files.**

| Species | Caller[1] | BND[2] | DEL[2] | DUP[2] | DUP:I[2,3] | INS[2] | INV[2] | Total[2] |
|---------|-----------|--------|--------|--------|------------|--------|--------|----------|
| *C. elegans* | pbsv | 2 (0) | 87 (11) | 22 (12) | NA | 115 (63) | 1 (0) | 227 (86) |
| | Sniffles | 0 (0) | 0 (0) | 0 (0) | NA | 34 (34) | 0 (0) | 34 (34) |
| | SVIM | 0 (0) | 15 (8) | 0 (0) | 0 (0) | 306 (293) | 0 (0) | 321 (301) |
| *A. thaliana* | pbsv | 26 (2) | 66 (12) | 19 (13) | NA | 149 (83) | 4 (0) | 264 (110) |
| | Sniffles | 0 (0) | 0 (0) | 0 (0) | NA | 58 (58) | 0 (0) | 58 (58) |
| | SVIM | 0 (0) | 79 (32) | 10 (8) | 0 (0) | 512 (474) | 0 (0) | 601 (515) |

Notes:
1. SV call sets were generated from Minimap2 aligned and SAMtools sorted BAM files (20X depth).
2. Values represent the global mean of the symmetric differences between the original and randomized FASTQ files for each strain or ecotype. Values outside parentheses describe the VCF level differences. Values inside parentheses describe the coordinate level differences.
3. The identification of interspersed duplications was only supported by SVIM.

majority of differences in *C. elegans* and *A. thaliana*, respectively. Although the total differences were highest in SVIM compared to pbsv and Sniffles, it generated the lowest overall Jaccard distances (Table S1), indicating that fewer differences were observed relative to the total number of SVIM calls. For each caller, fewer differences were observed when the comparison was limited to the SV type, VCF filter category (*e.g.*, PASS), and genomic coordinates for each SV call (Table 1). While less stringent than an exact comparison of all VCF fields, these criteria may be of interest to the analyst that does not perform further *post hoc* quality assurance following SV calling. Unsurprisingly, these criteria resulted in fewer call set differences for each caller (Table 1), with the overall differences again being highest in SVIM.

A wide range of differences were observed among the *C. elegans* strains (Fig. 1; Tables S2 and S3), suggesting that divergence from the reference strain may impact the sensitivity to FASTQ read order. Notably, the differences seen in XZ1516, the most divergent *C. elegans* strain (*Lee et al., 2021*), were among the highest observed for the strains within each call set. However, when the Jaccard distance was used to quantify read order sensitivity, this was only observed for SVIM. In pbsv, N2, the *C. elegans* reference strain, accounted for a significantly higher Jaccard distance in contrast to Sniffles and SVIM.

In contrast to *C. elegans*, fewer differences were observed among the *A. thaliana* ecotypes and similar patterns were observed for both the symmetric differences and Jaccard distances (Fig. 2; Tables S4 and S5).

## FASTQ read order sensitivity is affected by the alignment program

To evaluate the impact of the sequence aligner on FASTQ read order sensitivity, three programs were used to align the original and randomized FASTQ files prior to SV calling: Minimap2, NGMLR and pbmm2. In *C. elegans*, the highest overall Jaccard distance resulted from Minimap2 aligned reads in pbsv (0.022). Similarly, higher distances were generated in pbsv using the pbmm2 (0.016) and NGMLR (0.014) alignments compared those derived from other aligner and caller combinations (Fig. 3). For Sniffles, NGMLR generated the highest overall distance (0.005), followed by Minimap2 (0.004), and pbmm2
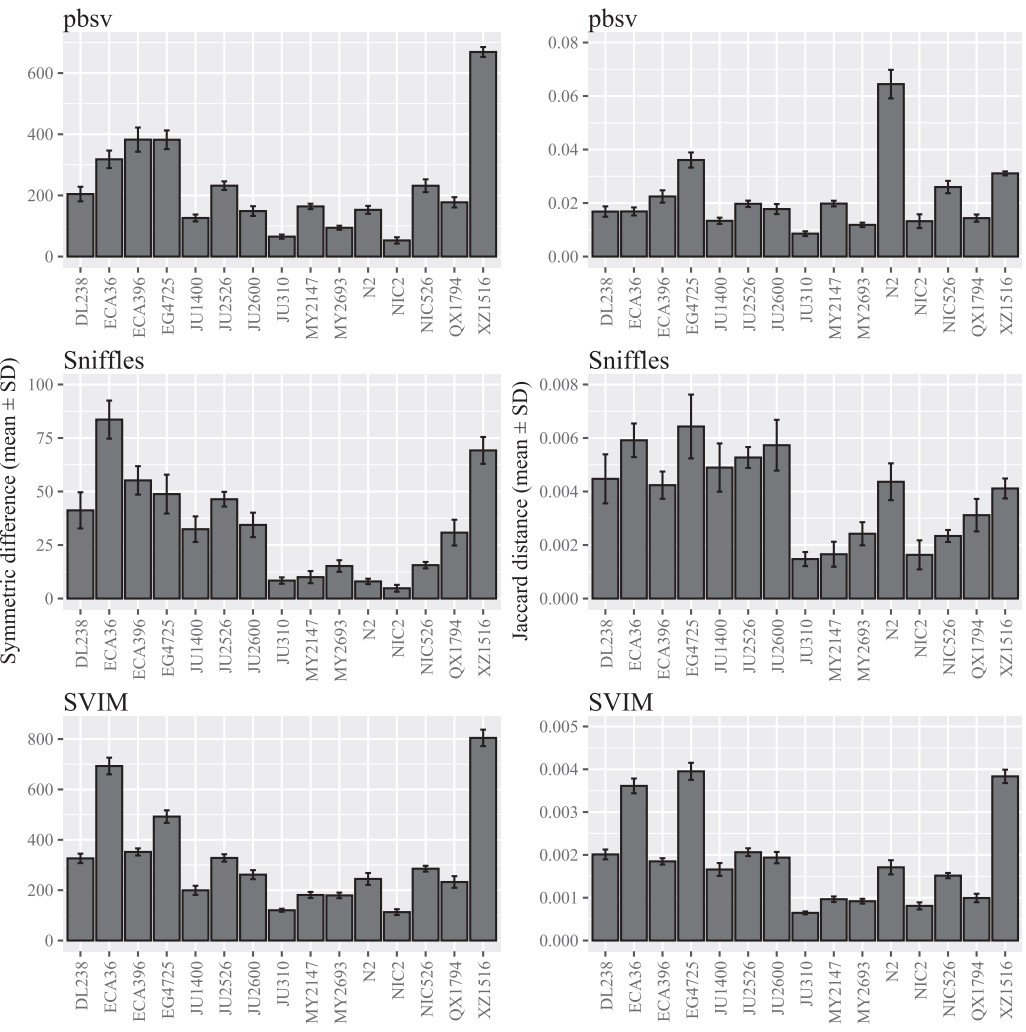

**Figure 1 SV call set differences in *C. elegans* attributable to FASTQ read order.** For each strain, five files with randomized read orders were created from the original FASTQ file. Each FASTQ file was subsampled to ensure that the Minimap2 alignment depths were 20X and the BAM files were sorted using SAMtools. The symmetric differences describe the disagreement between the call sets generated from the original and randomized FASTQ files. The Jaccard distances describe the proportion of predictions that disagree between the two call sets.

(0.002). Minimap2 generated the highest distance in SVIM (0.002), followed by NGMLR (0.001) and pbmm2 (0.001). No consistent pattern was observed for the impact of each aligner on read order sensitivity, as the relative order of the Jaccard distances differed between callers, as well as among the different strains (Figs. S1–S3).

In *A. thaliana*, pbsv also generated the highest overall Jaccard distances compared to the other callers and, similar to the *C. elegans* results, the differences were proportionally higher for the Minimap2 (0.017), pbmm2 (0.013) and NGMLR (0.010) alignments (Fig. 4). For Sniffles, the Minimap2 aligned reads generated the highest overall distance (0.006), followed by NGMLR (0.005), and pbmm2 (0.003). The highest overall distance in SVIM also resulted from Minimap2 alignments (0.005), followed by NGMLR (0.003) and

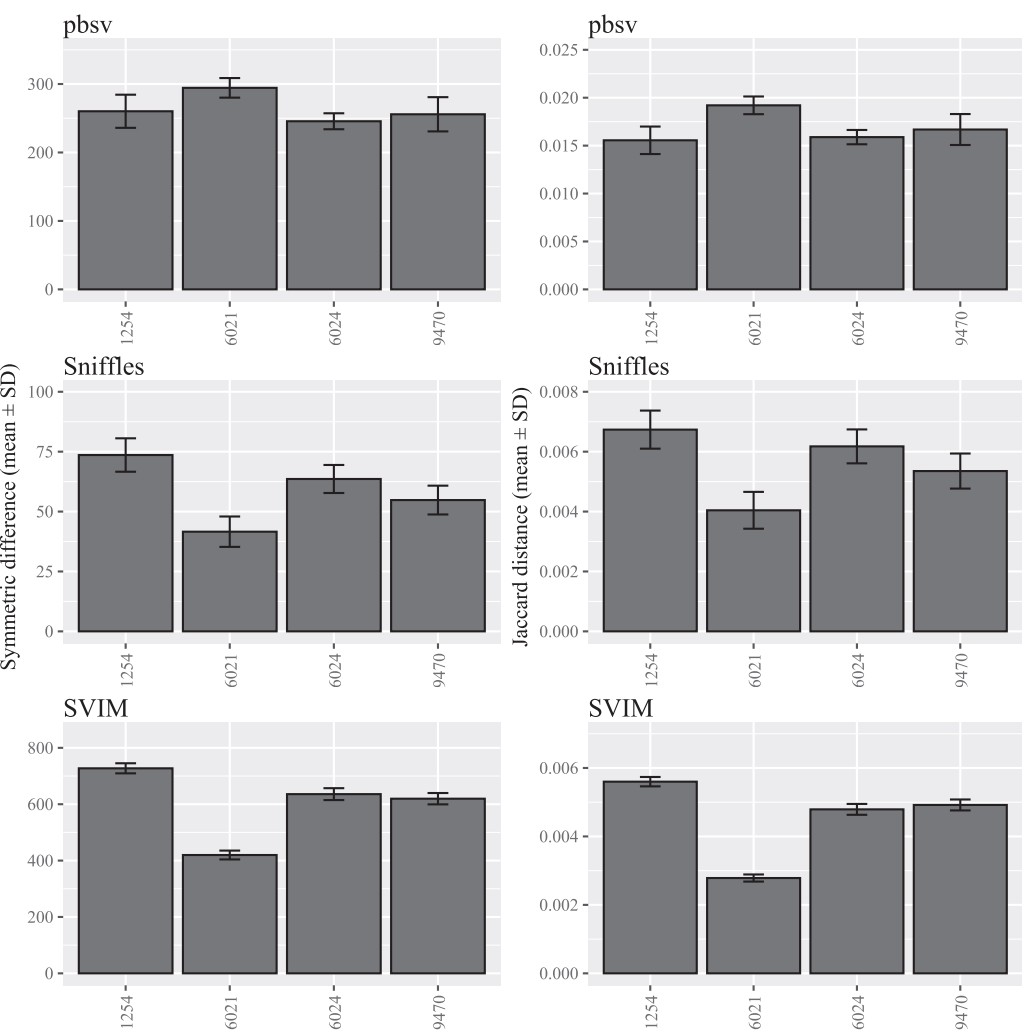

**Figure 2 SV call set differences in *A. thaliana* attributable to FASTQ read order.** For each strain, five files with randomized read orders were created from the original FASTQ file. Each FASTQ file was subsampled to ensure that the Minimap2 alignment depths were 20X and the BAM files were sorted using SAMtools. The symmetric differences describe the disagreement between the call sets generated from the original and randomized FASTQ files. The Jaccard distances describe the proportion of predictions that disagree between the two call sets.                

pbmm2 (0.003). For all ecotypes, Minimap2 generated the highest distances, whereas the relative impact of NGMLR and pbmm2 varied by strain and caller (Figs. S4–S6).

## Sequencing depth affects FASTQ read order sensitivity

To evaluate the impact of sequencing depth on read order sensitivity, the *C. elegans* FASTQ files were subsampled to ensure that four read depths (10X, 20X, 40X, and 60X) were obtained from each aligner. For each caller, the Jaccard distances increased at higher depths from the Minimap2 alignments (Fig. 5), a pattern also observed for NGMLR (Fig. S7) and pbmm2 (Fig. S8) aligned files. Notably, the pbsv Jaccard distances increased substantially with depth, which increased to 0.713 for the Minimap2 aligned data. In
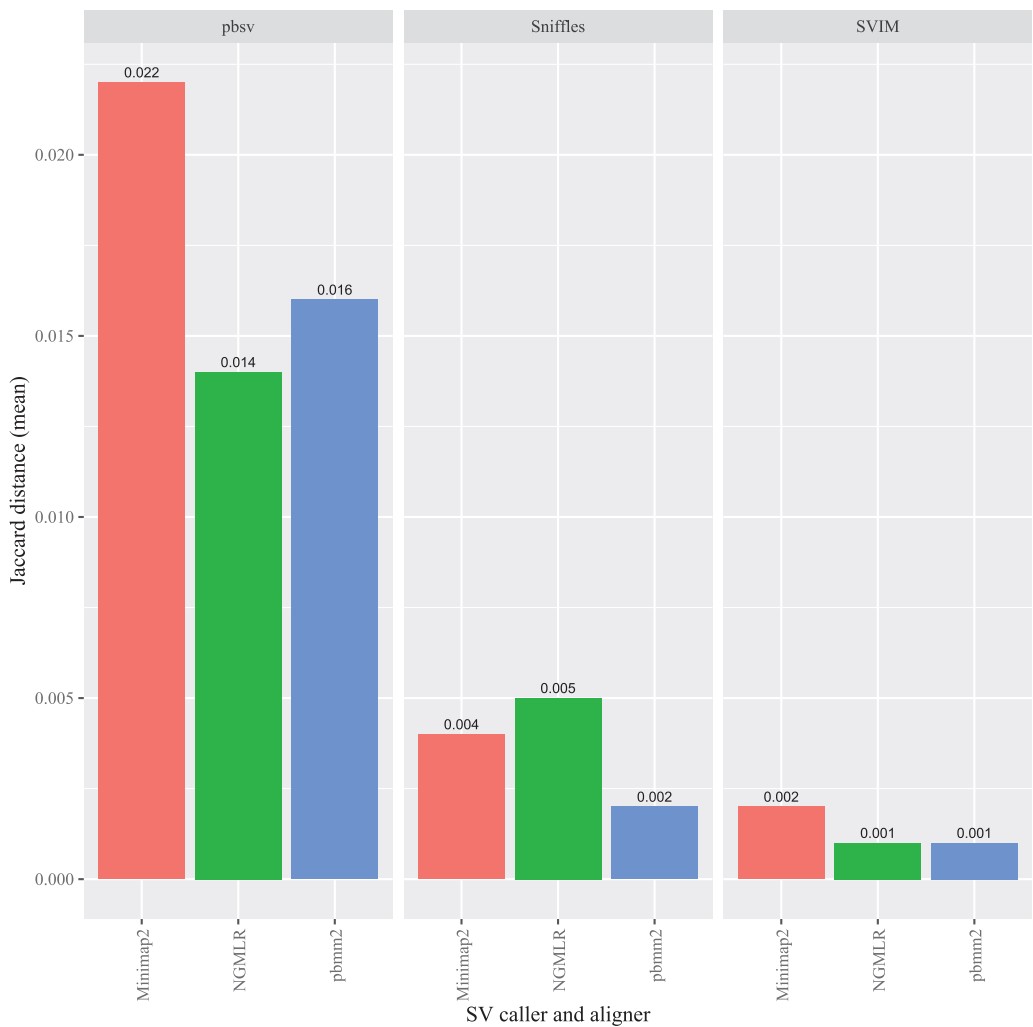

**Figure 3  Impact of the aligner on read order sensitivity in *C. elegans*.** For each strain, five files with randomized read orders were created from the original FASTQ file. Each FASTQ file was subsampled to ensure that the alignment depths of each aligner were 20X and the BAM files were sorted using SAMtools. The Jaccard distances describe the proportion of predictions that were in disagreement between the call sets generated from the original and randomized FASTQ files.

contrast, at full depth, the distances for Sniffles and SVIM only reached 0.025 and 0.031, respectively.

## BAM file sorting contributes to FASTQ read order sensitivity in SV calling

Several analyses were included to identify potential causes of the differences observed following the randomization of FASTQ file read order. Variant calling was repeated using the same alignment files, but identical results were obtained from each alignment, indicating that the callers are deterministic (*i.e.*, the same results are obtained given identical input data). For each strain or ecotype, the alignments generated from the original and permuted FASTQ files were compared. For each read alignment in the

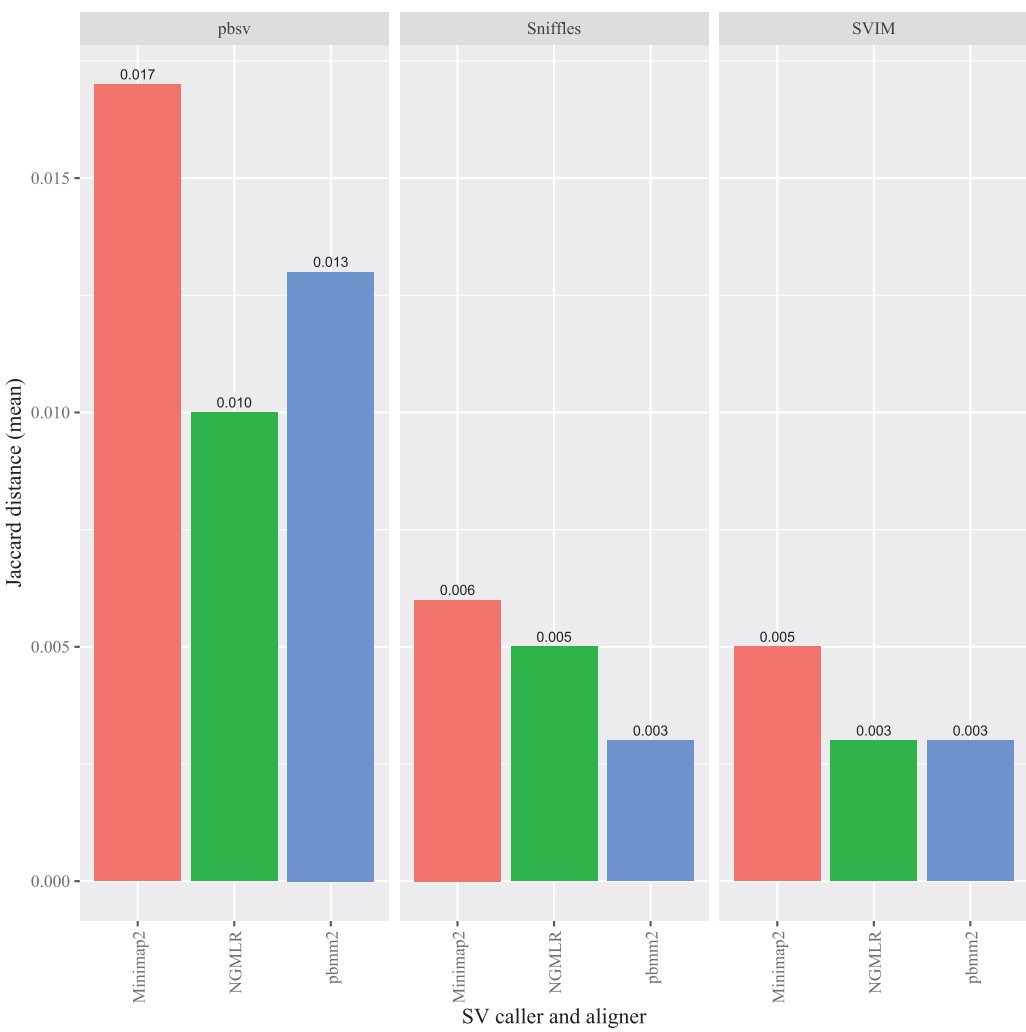

**Figure 4 Impact of the aligner on read order sensitivity in *A. thaliana*.** For each strain, five files with randomized read orders were created from the original FASTQ file. Each FASTQ file was subsampled to ensure that the alignment depths of each aligner were 20X and the BAM files were sorted using SAMtools. The Jaccard distances describe the proportion of predictions that were in disagreement between the call sets generated from the original and randomized FASTQ files.

original BAM file, identical alignments were present in the permutated versions. However, despite being sorted using SAMtools, the order of aligned sequences differed between the different BAM files. Further examination revealed that the order of alignments varied for sequences aligned to the same leftmost genomic coordinate, which is also described in the SAMtools documentation (http://www.htslib.org/doc/samtools-sort.html). Identical results were obtained from the original and permutated FASTQ files when they were sorted using Picard.

## DISCUSSION

Our results demonstrate the importance of considering the impact of FASTQ read order on SV calling and further highlight the need to consider how routine intermediate

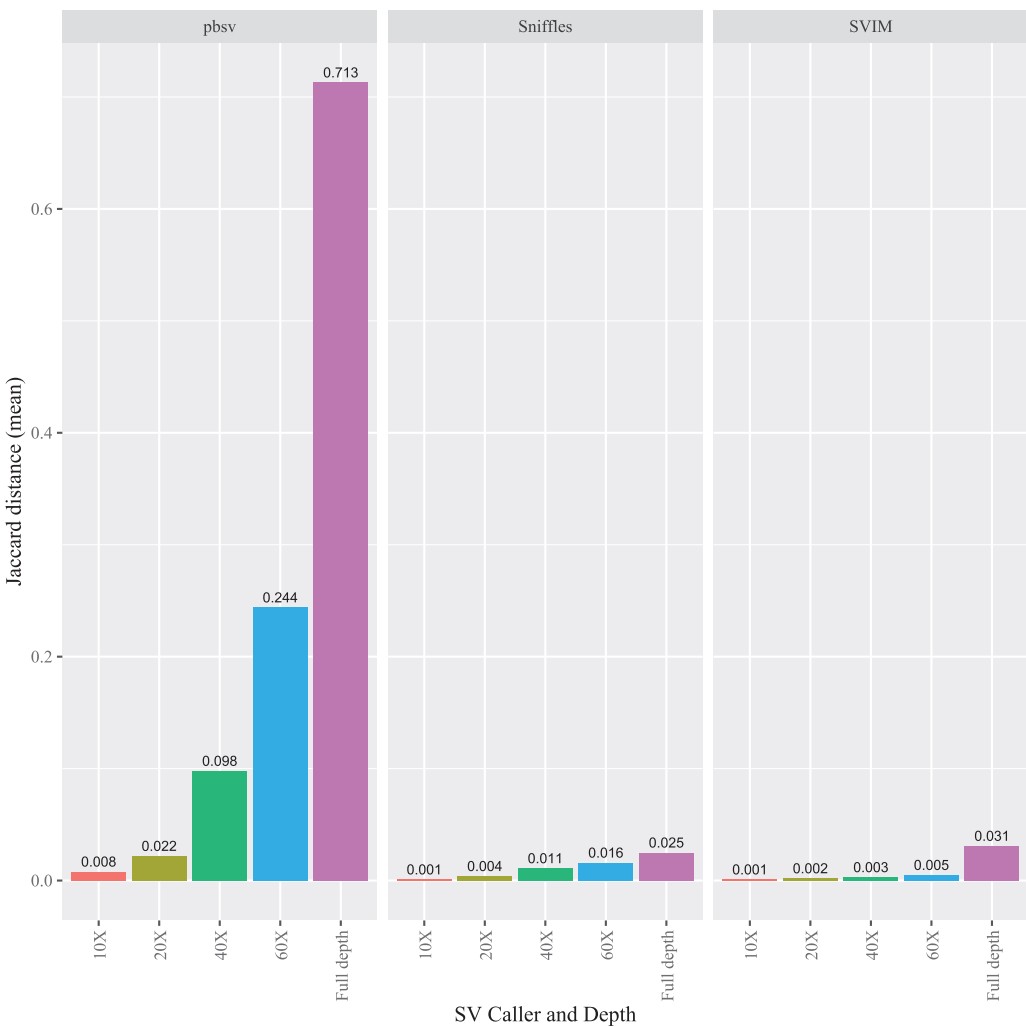

**Figure 5 Impact of sequencing depth on read order sensitivity in *C. elegans*.** For each strain, five files with randomized read orders were created from the original FASTQ file and subsampled to 10X, 20X, 40X, and 60X depth. The subsampled and full depth FASTQ files were aligned in Minimap2 and, subsequently, sorted using SAMtools. The Jaccard distances describe the proportion of predictions that were in disagreement between the call sets generated from the original and randomized FASTQ files.

methods, such as BAM sorting, can affect the results of a bioinformatics analysis. While the FASTQ file read order affected each caller, this sensitivity was considerably higher in pbsv, especially at higher read depths. Strikingly, over 70% of the pbsv SV calls differed when generated from the full depth Minimap2 alignments. Read order sensitivity also increased in Sniffles and SVIM at higher depths, albeit at considerably lower levels. As expected, fewer differences were observed when the SV call comparisons were limited to the SV type, VCF filter category, and genomic coordinates. Nonetheless, these findings have important implications for SV research, even if stringent *post hoc* quality assurance following SV calling is not a requirement. For example, even when using these relaxed criteria, 5% of the duplications called by pbsv from the 20X Minimap2 aligned *A. thaliana* data were in disagreement. The variability seen among the *C. elegans* strains raises the possibility that

genetic diversity within a species may contribute to read order sensitivity. In particular, the N2 reference strain accounted for the highest proportion of differences in pbsv. Given that most *C. elegans* research uses N2, these results should be considered for future laboratory work aimed at exploring structural variants in *C. elegans*.

Multiple factors must be considered to estimate the optimal sequencing depth for SV calling. These include desired recall, breakpoint accuracy, sequencing platform, tool choice, and the SV types under study. Generally, this choice involves trade-offs between accuracy and cost. With this in mind, we used subsampled data to determine if read order sensitivity is affected by sequencing depth. Although the read order sensitivity increased with higher depths for each caller, these differences were considerably higher for pbsv, where an overall Jaccard distance of 0.713 was attained using the full depth data. While the read order sensitivity may be discounted as an artifact of using exceptionally high read depths, high Jaccard distances were also observed in pbsv at 40X (0.098) and 60X (0.244). These depths are not drastically different from the higher range of recommendations from past benchmarks (typically between 15X and 30X) (*Sedlazeck et al., 2018*; *Liu et al., 2022b*; *Chen et al., 2023*) and are well below those that have been generated for other species, such as *Drosophila melanogaster* (*Rech et al., 2022*) and *Plasmodium knowlesi* (*Lapp et al., 2018*).

We explored several potential sources of variability associated with randomizing the FASTQ read orders. Repeated variant calling on the same input data generated identical SV calls, indicating that nondeterministic SV caller behaviour was not a source of the observed variability. Comparisons between the BAM files created from the FASTQ files with the original and randomized read orders indicated that each file contained identical alignments, but with occasional differences in their order. The algorithm used by SAMtools to sort the aligned reads was identified as the cause of the BAM file read order differences following randomization and, as a result, a factor in SV call variability due to input order sensitivity. Because SAMtools only uses the leftmost genome coordinate for sorting, permutation invariance cannot be guaranteed if multiple reads map to the same breakpoint. The analyst may therefore choose to sort the unsorted BAM files using another tool, such as Picard. Although this can ensure that the FASTQ read order does not affect the predicted SVs, it does not address the issue of input order sensitivity at the caller level. The sources of read order sensitivity were not identified for the SV callers, as a full exploration of all possible sources of variability would require analysis and modifications of the source code of each tool, which was beyond the scope of this study. Furthermore, this analysis would not have been possible for pbsv, as it is closed source and distributed as a binary file. It should also be noted that only reads mapped to the same leftmost coordinate would be subject to input order sensitivity when sorted using SAMtools. Therefore, these results should not be treated as an overall measure of the input order sensitivity of each SV caller.

Although a systematic evaluation of the sources of variability in each SV caller was not undertaken, the sensitivity to FASTQ read order can likely be attributed to how each caller clusters SV signatures to support an SV call. In a typical SV caller, variant signatures are first identified from the aligned reads and are subsequently clustered together (*Ho, Urban & Mills, 2020*). The signatures within a cluster can then be used to determine if enough

read support exists for a putative variant and to predict the SV breakpoints (*Guan & Sung, 2016*). To date, a range of clustering algorithms have been evaluated for SV calling, but considerable work remains to be done to identify the best approaches for clustering the full range of SV types and sizes. While each cluster would ideally be comprised of the total set of signatures supporting a single true variant, clustering is challenging in practice. For example, breakpoint imprecision has proved to be a major problem for optimal clustering, as the SV signatures for the same variant can vary due to sequencing and alignment errors (*Heller & Vingron, 2019*). Because SVs often occur in genomic hotspots, discriminating between signatures representing the same SVs from others in proximity can be difficult. Complex variants containing nested SVs add further challenges (*Ho, Urban & Mills, 2020*).

SV callers often include parameters that affect which signatures are clustered together, such as the maximum distances for variant size or genomic position. The number of reads supporting a cluster is typically used to assign a confidence score to the cluster, which is then used to determine the final SV call set (*Ritz et al., 2014*). Past benchmarks have demonstrated that read support and clustering stringency can have a major impact on SV calling and reinforce the importance of considering the trade-offs between precision and recall when selecting these parameter settings (*Korbel et al., 2009*). While these benchmarks provide valuable information on how read support and clustering parameter settings impact the accuracy of SV calling, the variability due to the input order sensitivity of clustering algorithms is underappreciated. Each caller in this study performs clustering to identify SV signatures that are expected to represent the same SV. In SVIM, SV signatures are grouped together using hierarchical agglomerative clustering (HAC), a bottom-up approach, where individual objects are first assigned to individual clusters that are successively merged based on their similarity (*Van Der Kloot, Spaans & Heiser, 2005*). This process is iterated until all objects are contained in a single cluster. A major problem in HAC occurs during the merging process when the minimum distance between multiple clusters is equal. Because these ties are usually resolved arbitrarily, HAC methods may yield different results after permutating the order of the input data (*Van Der Kloot, Spaans & Heiser, 2005*). In Sniffles, SV signatures are first assigned to bins according to their genomic coordinates. Neighboring bins are then merged based on a distance threshold derived from the starting coordinates of the signatures within each bin. These clusters are subsequently split to separate candidate SVs of different sizes (*Smolka et al., 2024*).

The analyses performed in this study were chosen to quantify variability in SV calling attributable to FASTQ and alignment read order and it is unclear whether SV callers that are more susceptible to changes in read order are more likely to be incorrect. Nonetheless, accuracy should be a concern for callers with higher sensitivity to input data order. In the most conspicuous case, permutating the read order could cause a given SV call to be present in one call set and absent from the other. Depending on whether the SV represents a true variant, it would either be a false positive or a false negative in one of the call sets. Variability in the breakpoint coordinates can also have important consequences, especially if the coordinates are predicted to result in high impact mutations, such as changes to protein coding sequences, splice sites, or regulatory elements. Although these analyses were not chosen to quantify caller accuracy, they may assist developers in the development

of more accurate tools. Benchmarks produced using the same input data for a given truth dataset may provide misleading estimates if the tools under evaluation are sensitive to read order. This is a plausible concern, as the limited availability of datasets with known SVs has resulted in most callers being benchmarked using the same data. Theoretically, if a tool is highly sensitive to read order, it may be optimized for the data used to benchmark its performance. By minimizing the read order sensitivity, the developer may be able to provide improved estimates of the tool's performance on other datasets.

Given the widespread use of clustering algorithms in SV callers, their impact on variant calling warrants further research. Key considerations also include how read order sensitivity is related to SV calling accuracy and if these predictions could be improved with other clustering approaches. Variability associated with the initial order of input data has been observed in a range of applications that involve clustering (*Jakobsson & Rosenberg, 2007*; *Boyce, Sievers & Higgins, 2015*; *Westcott & Schloss, 2015*) and a variety of methods have been developed to either reduce or eliminate its impact. For example, additional criteria may be used to decide which clusters should be merged when ties arise during HAC or to select a solution from repeated analyses of permutated input data. Tool developers may choose to evaluate similar approaches for clustering SV signatures, but their success would depend on finding suitable criteria to break ties or the feasibility of identifying a best solution from multiple analyses. It may also prove to be beneficial to reconsider the requirement of sorting read alignments prior to SV calling to assess its impact on clustering and to allow methods based on permutating the entire input data to be evaluated.

Clustering algorithms commonly used to assign amplicons to operational taxonomic units (*Müller & Nebel, 2018*) and to analyze gene expression data (*Oyelade et al., 2016*) have also been criticized for their sensitivity to the order of the input data they are provided. These concerns have led to the adoption of permutation invariant methods (*Mahé et al., 2014*; *Oyelade et al., 2016*; *Müller & Nebel, 2018*), which could also be evaluated for clustering SV signatures. In fact, one recent method for copy number variant calling (*Guo et al., 2022*) is based on WaveCluster (*Sheikholeslami, Chatterjee & Zhang, 2000*), a clustering algorithm that is insensitive to the order of input data. Further benchmarking will be necessary to quantify the impact of read order on SV calling and in the evaluation of different clustering approaches. Because different clustering approaches have their own strengths and weaknesses, these benchmarks would provide valuable information on the performance of different methods and the trade-offs between accuracy and speed. A better understanding of the relationship between read order sensitivity and accuracy would also allow tool developers to decide if improved clustering should be prioritized over addressing other sources of error.

Long-read sequencing has the potential to overcome some of the limitations of short-read approaches, but room for improvement remains for the current generation of callers. As tool developers work to improve algorithm performance, awareness of the effect of FASTQ read order on SV calling would be beneficial. This is of particular importance because only a small number of datasets with known SVs are currently available for benchmarking new tools. Although the results of this work should not be interpreted as

direct measures of accuracy, higher sensitivity on read order is cause for concern and should be included in discussions of reproducibility and replicability. Because the FASTQ file and alignment read order can affect the predicted variants, the randomness inherent in sequence order may contribute to failed replication attempts. Therefore, if future replication is a concern, we recommend that researchers sort alignment files using a tool that is insensitive to read order, such as Picard.

## CONCLUSIONS

Although many researchers are aware of the limitations of published benchmarks in bioinformatics, it is likely that differences resulting from random, arbitrary changes to the order of input data are underappreciated. Our results indicate that randomly permutating the order of reads in a FASTQ file can have a profound impact on the predicted structural variants. Seeing that the order of reads in a FASTQ file have no biological significance, we anticipate that our results will be of interest to tool developers interested in improving SV prediction. By quantifying the impact of read order, a developer may gain a better understanding of how random chance affects the relationship between the input data provided to their algorithm and the output it provides.

## ACKNOWLEDGEMENTS

We thank the University of Calgary's Faculty of Veterinary Medicine and Research Computing Services for their investment in and maintenance of high-performance computing facilities.

### Funding

This work was supported by a Discovery Grant (#04589-2020) from the Natural Sciences and Engineering Research Council of Canada (NSERC) to James D. Wasmuth. The funders had no role in study design, data collection and analysis, decision to publish, or preparation of the manuscript.

### Grant Disclosures

The following grant information was disclosed by the authors:
Discovery Grant: #04589-2020 from the Natural Sciences and Engineering Research Council of Canada.

### Competing Interests

The authors declare that they have no competing interests.

### Author Contributions

- Kyle J. Lesack conceived and designed the experiments, performed the experiments, analyzed the data, prepared figures and/or tables, authored or reviewed drafts of the article, and approved the final draft.

- James D. Wasmuth conceived and designed the experiments, authored or reviewed drafts of the article, and approved the final draft.

## Data Availability

The code is available at GitHub and Zenodo:

- https://github.com/kyleLesack/pacbio_read_order_shuffling

- kyleLesack. (2024). kyleLesack/pacbio_read_order_shuffling: v.1.0.0 (v1.0.0). Zenodo. https://doi.org/10.5281/zenodo.10611909.

PacBio sequencing data for the 14 *C. elegans* strains (DL238, ECA36, ECA396, EG4725, JU1400, JU2526, JU2600, JU310, MY2147, MY2693, NIC2, NIC526, QX1794, and XZ1516) are available from the *Caenorhabditis elegans* Natural Diversity Resource (CeNDR) database.

One sequencing run of the reference strain, N2 is available at SRA: DRR142768.

PacBio sequencing data for four *A. thaliana* ecotypes are available from the 1001 Genomes Project: 1254, 6021, 6024, and 9470; BioProject accession PRJNA779205.

## Supplemental Information

Supplemental information for this article can be found online at http://dx.doi.org/10.7717/peerj.17101#supplemental-information.

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
