# Peer review of "The impact of FASTQ and alignment read order on structural variant calling from long-read sequencing data"

_PeerJ, doi:10.7717/peerj.17101_

## Round 0.1 · original submission · Major Revisions

Dear Dr. Wasmuth,

I have received comments from three experts in the field on your work. While your work is potentially very important to the field, some methodological issues need to be clarified. All three reviews raised important questions and please address them fully if you'd decide to submit a revised version.

Thank you for submitting your work to PeerJ.

Hao Chen

Reviewer 1 ·

Basic reporting

no comment

Experimental design

I am concerned about the lack of a GitHub repository (or any other platform for code sharing). Given the nature of this work, reproducibility is particularly crucial here. Ensuring reproducibility includes providing access (or instruction about how to access) to raw data and the exact code used in the analysis. For example, this would avoid questions like “How did the authors exactly compare the SV call sets?”. SV representation might be an issue when comparing the same SVs that are just differently represented.

Another concern I have is that the tools used in this study are relatively old, which raises questions about the result reproducibility with the latest versions.

The study would benefit from performing multiple read order permutations (e.g., 3, 5, 7, or more) and summarizing the variability of the differences.

Validity of the findings

The study only covers one species, so I wonder how the findings would translate to other organisms, particularly those with more repetitive genomes (like plants) if repetitiveness is one of the causes of the issue. Testing on more diverse datasets could help to potentially clarify the drivers of the problem.

The study would benefit from addressing how different technologies could introduce reproducibility issues. For instance, how does the read order affect SV calling when Oxford Nanopore reads are used? Moreover, statistics about the read quality and length of the datasets used are missing. How do those 2 metrics affect the read order issue?

How this study should practically help users and developers? The study doesn't give concrete guidance on what users should do to alleviate the issue or where developers should work to modify and improve their tools.

Additional comments

Lesack and Wasmuth highlight the dependency of structural variant (SV) calling on read order in the input FASTQ files. Their alert is important, considering the reproducibility issue of bioinformatic analyses. However, I have several concerns that should be addressed to make the study a valuable help to the scientific community.

The manuscript lacks important information about where the differences in the results are stemming from and how their study would help researchers:
- Are the different call sets coming from specific genomic regions or contexts? A study that authors also cite in their introduction (Firtina & Alkan, 2016) reports read order issues for short read SV calling due to different read mappings in repetitive regions. With long reads, this issue should be less prominent. If differences arise from repetitive regions of the genomes, it might be worthwhile to evaluate also Winnomap2 as an aligner, since it is reported to better handle read mapping in repetitive regions.
- See my comment about covering more species.
- In the strict comparison, where authors check if the VCF files are exactly the same (ID excluded), what is really changing? How different the SVs are? Are SVs entirely different or close, or are we looking at different representations of the same SVs?
- See my comment about how this study should help researchers.

Line 30-31: I strongly suggest elaborating on this point by providing practical advice to researchers and developers on how to deal with such issues.

Line 35: the standard/usual definition of structural variants describes events >= 50 bp. Any reason for using the 100 bp threshold?

Line 105: minimap2’s citation is misplaced.

Line 110: why were authors unable to call SVs in pbsv from the minimap2 and NGMLR alignment?

Line 112: a filter of 100bp was applied, instead of 50bp. What happens to SNPs and INDELs? It would be interesting to report briefly what happens also for non-SVs.

Line 113-114: have authors checked that the entire pipelines are deterministic, so going from the same FASTQ to the same VCF files? The determinism of the entire pipeline needs to be examined, not just the SV caller.

Line 122: what about applying strong quality filters on the SVs? Does it reduce the reproducibility issue? What about stratifying by genomic context (low complexity regions, low mappability regions, repetitiveness, GC context)? More information about where these differences are is needed.

Line 145: why is Sniffles more robust with respect to read order? Possible explanations?

Line 164-166: can authors elaborate more on why the order of alignments varied for sequences aligned to the same leftmost genomic coordinates? It is not clear what is the problem and where to read more about it. Maybe the authors used a too-old version of samtools (1.9 instead of the last 1.17)?

Line 236-237: what about simulations? They are not perfect, but they might give us insight into possible trends like “differences in the SV calls due to read order are enriched in false positives or true positives”.

239-240: how authors are helping the developers? It is not clear how this study is helping them. Moreover, to be helpful, it is fundamental to use the latest versions of the tools, but the authors used several old tools: pbmm2 1.7.0 instead of 1.12.0, minimap2 2.17 instead of 2.26, samtools 1.9 instead of 1.17, Picard 2.27.5 instead of 3.0.0, pbsv 2.8.0 instead of 2.9.0, sniffles 2.0.6 instead of 2.0.7.

251-252: are authors sure that samtools sort’s instability is not related to a too-old samtools version? This aspect needs to be clarified.

Figure 3: why are only pbsv results displayed?

Reviewer 2 ·

Basic reporting

The manuscript submitted by Kyle Lesack and James D Wasmuth present an analysis of the effect of reads order in FASTQ files on the structural variation (SV) calling. The authors collected long reads sequencing data from 15 isolates of Caenorhabditis elegans and performed SV calling using three different algorithms: pbsv, Sniffles, and SVIM. They randomly shuffled the order of the reads in the FASTQ files and re-performed the SV calling. They then compared the results with the one obtained from the original files and measured the total number of calls which are not shared between the two sets, observing a percentage of differences that ranged from 2 to 60 percent between the different algorithms. The authors claim that their findings demonstrate the impact of reads order in the FASTQ files on the prediction of structural variants. They also subsampled to four depths each FASTQ file, to evaluate the effect of sequencing depth on these differences.
I find these results interesting, but the analyses need to be strengthened with additional controls. Moreover, the methodology used is sometimes difficult to follow and must be more clearly explained.

Experimental design

The major concerns are listed as follows:

The biggest concern is related to the general experimental design. In particular, instead of using the original files as a reference and performing a single test on shuffled data, the authors could generate multiple shuffled datasets and compare the number of calls in agreement between all the different datasets, this comparison will provide a more robust support to their conclusions.
The authors should provide a more detailed and numerical description of the datasets.
In the DNA Sequencing Datasets and Read Order Randomization section of the Methods, the authors can provide a description of the script used to shuffle the sequences, and also specify if a seed was used to allow replicable analysis. It is superfluous to mention the other scripts that the authors didn’t use in the end.
In the Sequence Alignment, subsampling, and Structural Variant Prediction section of the Methods a discussion on the algorithm will be the best way to affirm its deterministic nature (a check can be made to control if that topic was discussed in the papers in which these algorithms are presented). A simple single repetition of the experiment is not the best way to confirm that. An explanation about why it was not possible to do so otherwise will be helpful.
Is there a particular reason why the authors decided to focus on Caenorhabditis elegans data? If so, the authors should add this discussion to the manuscript.
In the introduction, the authors provided a clear discussion about the advantages of using long reads sequencing techniques and regarding the problems of short reads sequencing techniques for calling SVs. The description of the roles of structural variants and their different types could be expanded.
The procedure described in “Comparison of Predicted Structural Variants” can be reformulated in a clearer way.
Throughout the text, the authors need to provide more numerical details to support their claims. To cite just an example (but please check all claims throughout the paper):
- line 209 - "considerably higher” can be directly stated numerically.

Some minor comments:

In Figure 3 the y axis could be limited to show in a better resolution the differences between the different sequencing depths.
A bar plot of the differences observed between the different classes of SVs calls on the original FASTQ files vs the shuffled control would help readers to have a more direct picture of these differences.
there are missing words within the text, e.g.:
row 184 – “discrepancies occurred for (0.244 ± 0.083)” -> “discrepancies occurred for duplications (?) (0.244 ± 0.083)”.
row 234 – “the dependence of SV callers FASTQ file read order” -> “the dependence of SV callers from the(?) FASTQ file read order”.

Validity of the findings

no comment

·

Basic reporting

4. It is not clear which BAM sorting methods were used in Figure 1 & 2 and Table 1 & 2. Clarifying this information would enhance the paper's clarity.

5. Regarding lines 167-168, it is not clear which two elements are being referred to as identical. The clarification of this point is crucial as it directly impacts the core findings of the paper. For example, it could be "The discrepancy between the sorted BAM files from the original and permuted FASTQ files using SAMtools was also observed when using Picard for sorting" or "The Picard-sorted BAM file from the original and permuted FASTQ files was found to be identical."

7. Line 168: “aligned using Picard”. Probably a typo?

Experimental design

6. (If you want to add a little more to the paper) Within the 15 samples, one of the samples is the reference. Compare it to the rest of the samples would be a good sanity-check and yield some useful insights. It is expected to have less SVs. Will it also have a very different intersecting to non-intersecting SV ratios?

Validity of the findings

2. Personally, I have some concerns regarding the data presented in the article to support the main finding that “the order of input data had a large impact on SV prediction” (Line 22, Line 142).
-a. For Table 1 and Figure 1, The criteria for matching/consistent VCF entries appear to be overly stringent (“All VCF fields, except for the variant ID, were required to be equal in both the original and permutated SV calls to be classified as intersecting.”). I don’t feel comfortable drawing any major conclusions based on this criteria of intersecting.
-b. (Con.) Even with this stringent criteria, I still don’t think the data sufficiently support the language enough. Figure 1 demonstrates that the effect is dependent on the SV caller and large only for pbsv. 
-c. The criteria used for Figure 2 and Table 2 is already stringent enough. And I would recommend basing all the major conclusions on this intersecting criteria.
-d. (Con.) This will make the finding less shocking and more of something like “the order of input data has some but limited impact on SV prediction.” And this will be more in-line with people’s expectation and thus less “ground-breaking”, but I think the results are just as valuable.
-e. I would recommend moving Table 1 and Figure 1 to supplementary, but I will leave that to your own discretion. 

3. Related to the last point, I think Figure 3 should use the matching criteria used in Figure 2 instead of Figure 1.

Additional comments

1. After some quick search, I find no existing publications on the effect of FASTQ read orders on SV calling for long-reads. I think this is a non-trivial question and an important sanity check. I’m glad someone is doing it and sharing the results. I do believe that these findings hold value for the field.

8. As a new reviewer, please consider my comments with your own discretion. Thanks a lot.

---

## Round 0.2 · accepted · Accept

Dear Dr. Wasmuth,

All three reviewers recommended accepting the revised manuscript. But please do note additional comments by reviewer 3; you are welcome to incorporate additional modifications/clarifications in your final version.

Congratulations.

Hao Chen

Reviewer 1 ·

Basic reporting

no comment

Experimental design

no comment

Validity of the findings

no comment

Reviewer 2 ·

Basic reporting

The manuscript submitted by Kyle Lesack and James D Wasmuth presents an analysis of the effect of reads order in FASTQ files on the structural variation calling.
In the revised version of the manuscript the authors clarify the points raised by myself and the other reviewers at the first round of the review.

Experimental design

The authors solved my main concern, which was related to the use of only a single test on shuffled data, by generating multiple shuffled FASTQ files.

Validity of the findings

Additionally, sharing the GitHub repository and adding the additional A.thaliana species to the analysis helped make the work reproducible and more robust respectively.

·

Basic reporting

- I do agree the text and the writing describing the work has been greatly improved.
- I highly recommend adding a figure to describe the overall workflow. (Flowchart style)

Experimental design

- Please make it more clear what type of PacBio read you are using here: HiFi or CLR.

Validity of the findings

- Problem for variant calling algorithms or problem for bam sorting algorithm. I think authors need more discussion on where the source of variation is. (My impression here is that there is only one solo contributor for all the differences we are seeing here. The Samtools BAM sorting algorithm. If this is true, this needs to be emphasized more.) The majority of the text is really focus on how different callers react to it.
- Related to the last point, is it true that if the Picard are used for BAM sorting, you will not be able to observe any difference between results from the original and randomized FASTQ file? If so, this is really important, and should be emphasized in the paper.